# ECNet: Efficient Convolutional Networks for Side Scan Sonar Image Segmentation

**DOI:** 10.3390/s19092009

**Published:** 2019-04-29

**Authors:** Meihan Wu, Qi Wang, Eric Rigall, Kaige Li, Wenbo Zhu, Bo He, Tianhong Yan

**Affiliations:** 1School of Information Science and Engineering, Ocean University of China, Qingdao, Shandong 266000, China; wumeihan@stu.ouc.edu.cn (M.W.); wangqi6177@stu.ouc.edu.cn (Q.W.); 4e3.rigall.eric@gmail.com (E.R.); lkg@stu.ouc.edu.cn (K.L.); zwb5437@stu.ouc.edu.cn (W.Z.); 2School of Mechanical and Electrical Engineering, China Jiliang University, Hangzhou 310018, China

**Keywords:** side scan sonar (SSS), semantic segmentation, imbalance classification, image-to-image prediction, fully convolutional neural networks, deeply-supervised nets

## Abstract

This paper presents a novel and practical convolutional neural network architecture to implement semantic segmentation for side scan sonar (SSS) image. As a widely used sensor for marine survey, SSS provides higher-resolution images of the seafloor and underwater target. However, for a large number of background pixels in SSS image, the imbalance classification remains an issue. What is more, the SSS images contain undesirable speckle noise and intensity inhomogeneity. We define and detail a network and training strategy that tackle these three important issues for SSS images segmentation. Our proposed method performs image-to-image prediction by leveraging fully convolutional neural networks and deeply-supervised nets. The architecture consists of an encoder network to capture context, a corresponding decoder network to restore full input-size resolution feature maps from low-resolution ones for pixel-wise classification and a single stream deep neural network with multiple side-outputs to optimize edge segmentation. We performed prediction time of our network on our dataset, implemented on a NVIDIA Jetson AGX Xavier, and compared it to other similar semantic segmentation networks. The experimental results show that the presented method for SSS image segmentation brings obvious advantages, and is applicable for real-time processing tasks.

## 1. Introduction

Side scan sonar (SSS), among the most common sensors used in ocean survey, can provide images of the seafloor and underwater target. Target detection based on SSS image has a great variety of applications in marine archaeological surveying [1], oceanic mapping [2], and underwater detection [3,4,5], in which the main task is SSS image segmentation.

Various methods for SSS image segmentation have been proposed, most of which are based on unsupervised segmentation methods, such as active contour model, clustering segmentation method and Markov random field (MRF) segmentation method, etc. Generally, the common techniques of SSS image segmentation include the clustering segmentation method and the Markov random field (MRF) segmentation method. Specifically, Celik T. et al. [6] utilized clustering algorithm for SSS image segmentation. Firstly, in their algorithm, the multiresolution representation of the input image was constructed using the undecimated discrete wavelet transform (UDWT), where the feature vectors were extracted. Secondly, principal component analysis (PCA) was used to reduce the dimension of each feature vector. Finally, k-means clustering was used to group feature vectors into disjoint clusters. No extra prior assumptions were supposed in this algorithm. However, if there is a serious imbalance of data volume or different variances of classes, the clustering effect of k-means is not satisfactory. In addition, others are based on Markov random field (MRF) [7] and hierarchical Markov random field (HMRF) [8,9]. These methods for SSS image segmentation can get more ideal results. However, these methods may be dependent on parameters to a large extent and are not efficient. Supervised methods are mostly used for SSS image classification, but rarely for SSS image segmentation. As an example, convolutional neural network (CNN), a very effective and popular supervised methods, is commonly used for SSS image classification [10,11,12]. There exist other methods for SSS image classification by using convolutional neural networks combined with traditional classifiers [13,14]. In [14], the authors extracted the nonlinear features of sonar images by convolutional neural networks, and then used support vector machine (SVM) to classify targets. And authors prove that there are significant improvements in target recognition and classification of sonar images by deep learning method compared with other feature extraction methods. In particular, there are few methods based on full convolution network for SSS image segmentation.

Recently, convolutional neural networks have derived advances in recognition tasks. These networks do not only make great progress on tasks for image classification [15,16,17,18,19], but also for semantic segmentation. In 2015, Long et al. [20] designed fully convolutional networks (FCN) for semantic segmentation, which could be trained end-to-end on pixel-wise prediction and outperform the previous best results. Since then, semantic segmentation methods [21,22,23,24,25,26] based on the fully convolutional network have been developed rapidly. However, most of them are about natural scenes image segmentation, biomedical image segmentation [27,28], and satellite image segmentation [29,30]. These previous researches motivated the development of fully convolutional networks used for SSS image segmentation in this paper.

Our network, ECNet, is an efficient convolutional network for pixel-wise SSS image segmentation. It is mainly driven by SSS image understanding applications which are able to model appearance, shape, and understand the spatial relationship between target and background. It works with smaller training data and yields more ideal results. Inspired by SegNet [23], the core of our network architecture is the encoder–decoder pair. In this network architecture, the encoder splits information into feature maps to learn rich hierarchical features, and the decoder maps these features into spatial categorization to perform pixel-wise segmentation. The encoder reduces the resolution of feature map, so the decoding method would have a decisive influence in the model performance in increasing the resolution back. Inspired from the work in U-Net [27], SegNet [23], and LinkNet [25], the appropriate decoding method was determined through a series of experiments.

Considering that the intermediate layers in networks are plentiful of useful fine details, side-output layers were added to the network to learn nested multiscale feature and effectively utilize features in intermediate layers. To solve the problem of imbalance classification and improve the network performance, we use weighted loss, where pixels predicted to be the targets would obtain a larger weight in the loss function. In experiments, the proposed segmentation network was compared to U-Net, SegNet, and LinkNet, and the experimental results demonstrate a good segmentation ability and generalization performance.

The remainder of this paper is organized as follows. We introduce related works on semantic segmentation of CNNs, the residual building blocks, and a single stream deep network using multiple side-outputs in Section 2. In Section 3, the network architecture and relevant formulation are described. In Section 4, we detail the experimental process, where we analyze the data and give some parameters for the experiment. Section 5 demonstrates experimental results and discusses the design choices. Finally, we give our conclusions in Section 6.

## 2. Related Works

Semantic segmentation, fueled by challenging datasets [31,32,33,34], is an active research topic. For sonar image segmentation, there are fewer methods using the fully convolutional network. In this paper, we rearchitect a network for dense prediction of pixel-wise SSS image segmentation. Our network architecture draws on recent promising results of fully convolutional networks for image segmentation and edge detection [35,36].

One of the strengths of CNN is that its multilayer structure can automatically learn multiple levels of features. These abstract features are used to decide the class of objects classes within an image, which is useful for categorization. Since some details of objects are lost, it is difficult to get the specific outline of objects and the category of each pixel. Hence, it is difficult to achieve image segmentation for CNN. This situation was improved until the idea of fully convolutional network appeared in Shelhamer et al. [20]. Pixel level prediction tasks achieve great progress by replacing fully connected layers by convolution layers and inserting differentiable interpolation layers and a spatial loss. Thus, fully convolutional networks, which showed effective feature extraction and end-to-end training, become one of the most popular solutions for semantic segmentation tasks.

Most architectures, particularly designed for segmentation, have an encoder network and a corresponding decoder network [23,25,27]. The former, learning rich hierarchical features of input data, is composed of classification networks’ modules, such as VGG [16], GoogLeNet [17], and ResNet [18]. These classification networks achieve good results for dealing with recognition tasks. Among them, ResNet have proved its superiority in several challenging recognition tasks, such as ImageNet [37] and MS COCO [38] competitions, and the number of semantic segmentation networks by means of residual building blocks is increasing. He K. et al. [39] investigated and analyzed the residual building blocks by a series of ablation experiments. These experiments, emphasizing the key role of skip connections and the importance of their identity mappings, supported the idea that keeping a “clean” information path is helpful for making the optimization easier. Based on these experiments, our encoder unit use “full preactivation unit”, proved to perform better than the baseline counterparts.

Deep learning, especially convolutional neural network (CNN), has active performance in image preprocessing because of its self-learning ability. In recent years, image denoising methods based on deep learning have also been proposed and developed. In 2008, Jain et al. proposed a denoising method based on CNN for processing natural images [40], which can yield similar or even better results than conventional methods, such as wavelet transform and Markov random field. In 2016, Mao et al. proposed a deep fully convolutional autoencoder network for image denoising [41]. In this network, the convolutional layer is responsible for feature extraction, capturing abstract information from the image content while eliminating image noise or useful information loss. Correspondingly, the deconvolution layer is used to restore image details.

The aforementioned models relying on fully convolutional networks have significantly contributed to the state-of-the-art techniques, but all of them still lose some useful features from middle layers. In order to utilize the characteristics of the middle layer to the best of one’s ability, we exploit the approach in holistically-nested edge detection [35]. Holistically-nested edge detection (HED) mainly solves two problems. One is the end-to-end training and prediction, and the other one is multiscale and multilevel feature learning. Through their architecture, the authors not only complete the prediction from image to image, but also perform edge detection by the learned rich hierarchical features from middle layers. A single stream deep network with multiple side-outputs was considered for multiscale learning, by which the authors highlight the importance of obtaining edge maps. The side-output layers were then interpolated behind encoder unit which encourages coherent contributions to improve the accuracy. It can generate predictions from multiple scales, but there is no significant redundancy of both representation and computational complexity.

## 3. Methods

### 3.1. Network Architecture

Our network architecture is illustrated in Figure 1. The main module of our network is shown in Figure 1b, where the side-output network are illustrated from top to bottom, the encoder network consisting of four encoders and the decoder network with the four corresponding decoders. Given an input image in Figure 1a, an encoder network made up of residual building blocks extracts the feature maps. Three side-output layers are connected to the last layer of encoder1–3 to get relevant side-outputs of different scales. In ECNet, the inputs of decoder1–3 are the sum of the corresponding encoder outputs and the previous decoder outputs. Each decoder takes advantage of the max-pooling indices, stored by the corresponding encoder during the max-pooling operation. The decoders use these indices to achieve up-sampling operation on their input feature maps. The inputs of decoder4 are only from encoder4. Then, the main outputs are obtained after decoder1. By doing this, spatial information lost during the encoder operations are recovered and coarse high-level feature maps can be refined. Finally, these three side-outputs and the main outputs are combined together, before being averaged to get the final prediction map in Figure 1d.

The architecture of an encoder unit is illustrated in Figure 2a. This encoder is composed of a residual block and a convolutional block. In order to enhance the expressive ability of the encoder network and to make the whole model easier to learn and optimize, each encoder unit uses skip connection referencing full preactivation unit [39]. Then, each residual block outputs are passed through one convolutional block to fit the layers dimension. Layers within encoder block are shown in Figure 2a. Here, conv means convolution with the kernel size of 3. MaxPooling represents down-sampling by a factor of 2, which is achieved by means of performing max-pooling operations. "Batch normalization (BN), followed by ReLU nonlinearity [42,43] is placed ahead of convolutional layer. Layer details for decoder unit are shown in Figure 2b. MaxUnpooling denotes the nonlinear up-sampling operation by the max-pooling indices received from the corresponding encoder. The decoder1 outputs are fed to a classifier to independently estimate class probabilities for each pixel, as seen in Figure 2, used as the main output. Table 1 contains the information about the feature maps used in each of these blocks.

In general, spatial information is lost in the encoder due to pooling or convolution operations. Therefore, the decoder should map low-resolution features to input resolution for pixel-wise classification. The decoding method thus plays a key role in the model performance. There are many kinds of existing encoding methods, such as U-Net [27], SegNet [23], and LinkNet [25]. U-Net decoding method creates an up-sampling stage according to the corresponding down-sampling stage in the original network. Feature fusion is then performed using channel connection with the corresponding stage. This one provides more perspective, but it leads to an increase in the number of channels affecting the computational efficiency. In SegNet, the decoding network uses the max-pooling indices stored and transmitted from the corresponding encoder for up-sampling to obtain a sparse feature map. This method has fewer parameters to fit. In LinkNet, the outputs of the encoder are directly added to the inputs of the corresponding decoder. The decoding method, bypassing spatial information, improves performance along with a significant decrease in processing time. In this way, information which would have been lost using other decoding methods is here retained, and no additional parameters and operation are required in learning this information.

Our decoding method leveraged the combination of the max-pooling indices from SegNet and direct connection from LinkNet. In this paper, experiments prove that our method provides more accurate segmentation results.

Each of the last layers in encoder1–3 is followed by a 1 × 1 conv layer in the side-output module. Then, a deconv layer up-samples the feature maps to get original image size back. Finally, a sigmoid layer is followed to get the outputs and loss in side-output layers. As encoder units have different sizes of receptive field, our network can learn multilevel information, in particular object-level and low-level, which is useful for semantic segmentation. Table 2 is a summary of the receptive fields and strides configuration. The outputs of each side-output layer are multiscale, where the smaller the side-input size, the larger the receptive field.

### 3.2. Formulation

We denote our input dataset by $D=\left\{\left(X_{n},Y_{n}\right),n=1,\ldots ,N\right\}$, where sample image $X_{n}=\left\{X_{ij}^{\left(n\right)};i=1,\ldots ,w,j=1,\ldots ,h\right\}$ denotes the raw input image with the width (*w*) and the length (*h*), and $Y_{n}=\left\{Y_{ij}^{\left(n\right)};i=1,\ldots ,w,j=1,\ldots ,h;Y_{ij}^{\left(n\right)}\in \left\{0,1\right\}\right\}$ represents the related ground truth binary map for image $X_{n}$.

Each encoder is composed of a residual block and a convolutional block. More concretely, the entire set of standard network layer parameters in residual block and a convolution block are represented as $w_{L}$ and *w*, respectively. The functions of convolution block compute outputs $y_{i}$ by
(1)$$y_{i}=F\left(x_{i},w_{i}\right)$$
where, *F* determines the layer type: A batch normalization layer, a ReLU nonlinearity for an activation function, and a matrix multiplication for convolution. Each residual block can be expressed in a general form:(2)$$X_{L+1}=X_{L}+\overset{2}{\underset{I=1}{\sum }}F\left(X_{L},w_{Li}\right)$$
where, $X_{L}$ and $X_{L+1}$ denote the respective inputs and outputs of the *L*-th residual block.

We represent the entire set of network layer parameters as Θ. Suppose there are M side-output layers in the network. Each of them is assigned with a classifier, so the corresponding weights can be defined as $\left(\theta^{\left(1\right)},\ldots ,\theta^{\left(M\right)}\right)⊑\Theta$. Since most areas of the image are the seabed reverberation areas and only a few are the object-highlight areas, the imbalance classification issue cannot be avoided in SSS image datasets. To solve this problem, the following class-balanced cross-entropy loss is computed in this paper. Loss function used in the main-output layer is as follows:(3)$$l_{main}\left(\Theta \right)=-\frac{1}{n}\left[\overset{n}{\underset{i=1}{\sum }}\left(\alpha y_{i}\log P\left(y_{i}=1|X;\Theta \right)+\left(1-\alpha \right)\left(1-y_{i}\right)\log P\left(y_{i}=0|X;\Theta \right)\right)\right]\;\;+\lambda R\left(\Theta \right)$$
where, *n* represents the number of training samples in each batch and $\alpha =\left|Y_{-}\right|/\left(\left|Y_{-}\right|+\left|Y_{+}\right|\right)$, $1-\alpha =\left|Y_{+}\right|/\left(\left|Y_{-}\right|+\left|Y_{+}\right|\right)$ corresponds to the ratio of background pixels over all the pixels. $\left|Y_{+}\right|$ and $\left|Y_{-}\right|$ represent the number of target pixels and background pixels from the ground truth, respectively. $P\left(y_{i}=0|X;\Theta \right)\in \left[0,1\right]$ is calculated using sigmoid function on the activation value at pixel i, in order to obtain the main-output map predictions ${\hat{Y}}_{main}$. R stands for the regularization term, and λ is the hyperparameter of the regularization. Similarly, the loss function of the side-output is represented as follows:(4)$$\begin{array}{ll} l_{side}^{\left(m\right)}\left(\Theta ,\theta^{\left(m\right)}\right)= & -\frac{1}{n}[\overset{n}{\underset{i=1}{\sum }}(\alpha y_{i}\log P\left(y_{i=1}|X;\Theta ,\theta^{\left(m\right)}\right)\\ & +\left(1-\alpha \right)\left(1-y_{i}\right)\log P\left(y_{i}=0|X;\Theta ,\theta^{\left(m\right)}\right))]+\lambda R\left(\Theta ,\theta^{\left(m\right)}\right). \end{array}$$

So, the following objective function is considered:(5)$$L\left(\Theta ,\theta^{\left(m\right)}\right)=\beta l_{main}\left(\theta \right)+\overset{M}{\underset{m=1}{\sum }}\beta_{m}l_{side}^{\left(m\right)}\left(\Theta ,\theta^{\left(m\right)}\right)$$
where, $\left(\beta ,\beta_{1},\ldots ,\beta_{m}\right)$ are some trade-off parameters introduced for loss function from both the main-output and side-output, which are represented here as a series of hyperparameters. Then, the objective function below is minimized by stochastic gradient descent:(6)$$\left(\Theta ,\theta^{\left(m\right)}\right)=argmin\left(L\left(\Theta ,\theta^{\left(m\right)}\right)\right).$$

Given an input image $X_{n}$, predictions are obtained from both the main output ${\hat{Y}}_{main}$ and side-output ${\hat{Y}}_{side}^{\left(m\right)}$. During testing, we can select main output ${\hat{Y}}_{main}$ or generated maps by further aggregating ${\hat{Y}}_{f}$ as final output. The ${\hat{Y}}_{f}$ is defined as follows:(7)$${\hat{Y}}_{f}=Average\left({\hat{Y}}_{main},{\hat{Y}}_{side}^{\left(1\right)},\ldots ,{\hat{Y}}_{side}^{\left(M\right)}\right)$$

## 4. Experiment

The experimental scheme of our SSS image segmentation process is provided in Figure 3. Accordingly, this section is arranged as follows: The collection method of dataset images is described and the SSS image preprocessing stage is shown in Section 4.1; the method used to get the ground truth of dataset is introduced in Section 4.2; the implementation details and parameters setup of the experiments are described in Section 4.3.

### 4.1. SSS Image Preprocessing 

SSS data are collected by dual-frequency side-scan sonar (Shark-S450D) in Fujian, China, where some of the data are in Figure 4. For the original SSS image, the proposed method can be used to get better performance with only some simple preprocessing. This also shortens the processing time. Therefore, we removed middle waterfall and performed bilinear interpolation on the raw SSS image. In Figure 5a, the results after these operations are provided.

### 4.2. Dataset

The ground truth for images is essential to use supervised method for SSS image segmentation. We used LabelMe, an image annotation tool developed by the MIT, to manually label images. In the corresponding ground truth image (shown in Figure 5b), the blue and the black color denote the object and the background areas, respectively.

The data image (shown in Figure 5a) and its corresponding labelled image (shown Figure 5b) were then cut at the size of 240 × 240 and with a stride of 50. We selected samples with target pixel ratio exceeding 5% to form a dataset. We selected 70% of its images as our training set, and the other 30% as our test set. In the training set, about 20% of the data were randomly selected as a validation set. More precisely, the dataset consisted of 3528 training images, 882 validation images, and 959 testing images.

Some challenging factors existed in the dataset, which are illustrated in Figure 6. Figure 6a illustrates a clear and standard SSS image. In Figure 6b, some target pixels are clear and others are weak. In Figure 6c, the images are dark. In Figure 6d, the target pixel areas are discontinuous. In Figure 6e, target pixels are weak and unclear. In Figure 6f, there is strong noise.

In addition to analyzing some challenging factors in the dataset, the number of target and background pixels in the dataset were also compared, shown in Figure 7. There is a clear serious category imbalance in the dataset, which brings huge challenges to the model performance.

### 4.3. Experimental Parameters

Models were all trained by a NVIDIA Quadro M5000 card, which provides sufficient computational power and resources to compute the model weights. Then, in order to ensure that these models can achieve real-time processing for on-board applications, the experimental tests concerning the prediction time were then performed on an embedded platform, NVIDIA Jetson AGX Xavier.

To implement our model on an NVIDIA Quadro M5000 card, stochastic gradient descent (SGD) was used with the global learning rate of 1e^−3^ and momentum of 0.9 and 0.999. We trained the variable parameters until the training loss converges using PyTorch [44]. The hyperparameters and the values were fixed as follows: Mini-batch size (8), weight decay (0.0002), loss-weight $\beta_{m}$ (1) for each side-output layer, and loss-weight β (1) for main-output layer. Prior to each epoch process, it was necessary to shuffle the training set and then take each mini-batch out of the epoch in order to ensure that each image was used only once in an epoch. After that, each image was normalized. The model which performed the best on the validation set was finally obtained.

## 5. Results and Discussion

We compared different models’ performances, using four common performance metrics [20]: Pixel accuracy (Pixel Acc.), measuring the proportion of pixels accurately predicted to the total pixels, mean accuracy (Mean Acc.) which is the average of the prediction accuracy over all categories, mean intersection over union (IU), and frequency weighted IU (f.w. IU). These last two metrics define the variations on region intersection over union (IU) used in target detection. IU is the overlap ratio between the candidate bound and the ground truth bound. In other words, IU is the ratio between the intersection and the union. The mean IU was calculated as follows:(8)$$U=\frac{\left|C\cap G\right|}{\left|C\cup G\right|}$$
where, |C∩G| corresponds to the intersection of candidate bound (*C*) and ground truth bound (*G*), and |C∪G| represents their union. Mean IU was calculated for each class and then averaged, both of which were calculated using the ground truth and the segmentation results. Frequency weighted IU is an elevation of mean IU, which can set weights for each class according to their relative frequency. The pixel accuracy (Pixel Acc.) was obtained using the equation below: (9)$$\mathrm{Pixel}\;\mathrm{Acc}=\sum_{i}u_{ii}/\sum_{i}\sum_{j}u_{ij}$$
the mean accuracy was calculated as follows:(10)$$\mathrm{Mean}\;\mathrm{Acc}=\left(1/k\right)\sum_{i}\frac{u_{ii}}{\sum_{j}u_{ij}}$$
the mean IU was obtained using the equation below:(11)$$\mathrm{mean}\;\mathrm{IU}=\left(1/k\right)\sum_{i}\frac{u_{ii}}{\sum_{j}u_{ij}+\sum_{j}u_{ji}-u_{ii}}$$
and the function of frequency weighted IU (f.w. IU) was:(12)$$f.w.\;\mathrm{IU}=\sum_{i}\left(\sum_{j}u_{ij}/\sum_{k}\sum_{j}u_{ij}\right)\frac{u_{ii}}{\sum_{j}u_{ij}+\sum_{j}u_{ji}-u_{ii}}$$
where, k represents the number of classes. The symbol $u_{ij}$ corresponds to the number of samples belonging to category i in ground truth and are classified in class j in segmentation results.

Our decoder unit is shown in Figure 2b, and a detailed comparative study of different decoding methods is provided in this section. In order to verify the proposed decoding method, we carried out the following experiment: Keeping the proposed network structure unchanged and testing different decoding methods. Table 3 shows their comparison results. Our decoding methods provide a more detailed and accurate segmentation than others in our architecture.

In the previous section, six challenging factors of SSS images were analyzed. Figure 8 shows the corresponding segmentation results. Generally speaking, the model we propose can yield better results. However, when the SSS image is too dark or the target pixel is very unclear, the segmentation of the SSS image performs worse than ideal. ECNet is also compared with overall architecture of U-Net, SegNet and LinkNet in our dataset. Table 4 lists the relative comparison results.

Compared with other networks, our proposed network reduces the receptive field. This could lead to a decrease in the model performance, because the maximum size of the targets able to be detected is thus reduced, potentially missing bigger targets. However, because the target area of the sonar data is small, it has a small impact on the sonar images, as proved by the experiments. As shown in Table 4, ECNet outperforms LinkNet and SegNet in four measures, while U-Net performs slightly better than ECNet in pixel accuracy, mean IU, and f.w. IU. LinkNet, SegNet, and U-Net all have larger receptive fields than ECNet, so that reducing the receptive field has a small impact on SSS images with smaller target areas. From this, we are inclined to the view that it is critical to adjust the model structure to make it easier to optimize, which may be one of the reasons why U-Net performs best. Our approach, reducing the number of parameters and model size, makes our model take up the less computing resources on an embedded platform, which is critical for being applied in real-time tasks.

The inference speed of ECNet and existing networks of semantic segmentation on NVIDIA Jetson AGX Xavier were reported. Our default image resolution was set to 240 × 240, and Table 5 reports the number of parameters, the size of the model, and the inference time for a single input. As indicated from the numbers provided, ECNet uses less memory and performs faster. Furthermore, it can provide real-time performance on NVIDIA Jetson AGX Xavier, which can be employed for real-time inspection application.

ECNet realizes the best trade-off between effectiveness and efficiency according to Table 4 and Table 5. Although U-Net has a slightly higher accuracy than our network, ECNet is much faster. Moreover, our model is much smaller than others. Since our model is simple and efficient, it can be easily applied in various tasks.

The segmentation of the SSS image is less ideal when the SSS image is too dark or the target pixel is very unclear. In a further study, authors can add an image enhancement process for SSS images, such as increasing image brightness, adjusting image contrast, and histogram equalization. The network performance can be improved through data augmentation, such as rotating the images to different angles and cropping the largest rectangle in them, or flipping images at each angle. It is worth mentioning that our model and strategy, based on HED approach, still have not clearly considered neighboring pixels information. In the future, authors would like to explore how to optimize the model using the context between neighboring pixels.

## 6. Conclusions

In this paper, we presented a novel neural network architecture designed for semantic segmentation of SSS images, where a novel encoder was designed to learn rich hierarchical features. Additionally, we took advantage of a single stream deep network using side outputs after each encoder to learn useful features from middle layers. To consider the influence of the imbalance classification problem, the model made use of weighted loss, where the target pixels resulted in a larger weight in the loss function. Finally, ablation studies were performed to compare different decoding methods. These show that skip architecture in our decoding method provides the best compromise between computational efficiency and accuracy.

ECNet allows to perform predictions much faster and more efficiently, which makes it possible to effectively utilize the limited resources on embedded platforms. There is no doubt that the ECNet can be more broadly applied to real-time tasks.

## Figures and Tables

**Figure 1 sensors-19-02009-f001:**
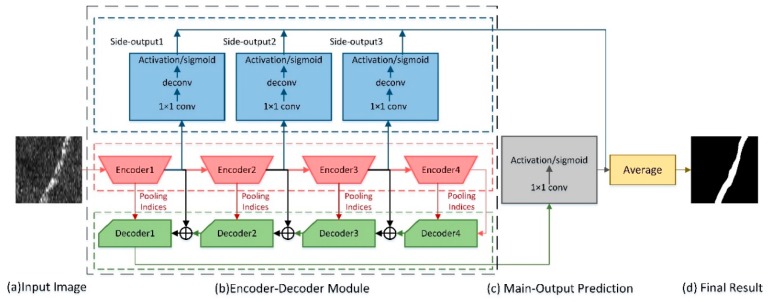
An illustration of the ECNet architecture. Three side-output layers are placed behind encoder1-3. Deep supervision is performed on each side-output layer, conducting the side-outputs to the predictions we expect. Our encoder–decoder module has no fully connected layers in encoder–decoder module. In addition to the inputs of the decoder4, the inputs of the other decoders are the sum of the corresponding encoder outputs and the latest decoder outputs. A decoder performs an up-sampling operation on its input based on the pooling indices the corresponding encoder provide. The feature maps of the final decoder output are given as inputs to a sigmoid classifier to complete the pixel level classification.

**Figure 2 sensors-19-02009-f002:**
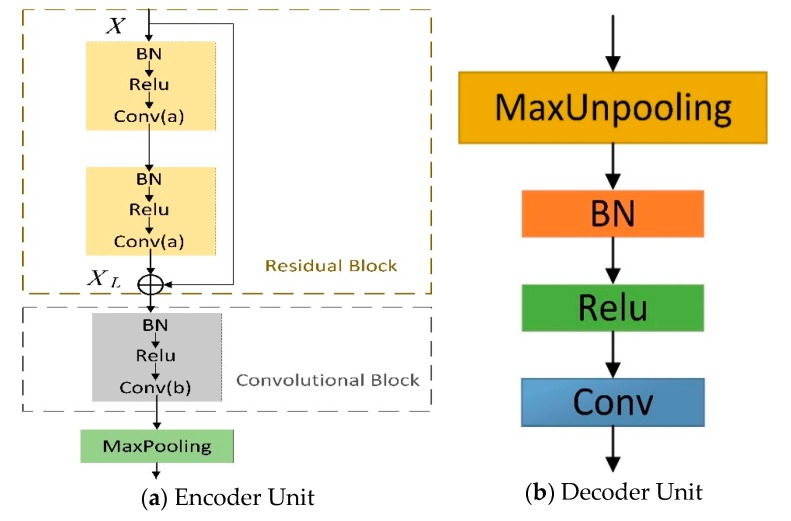
The architecture of an encoder unit (**a**) and a decoder unit (**b**).

**Figure 3 sensors-19-02009-f003:**
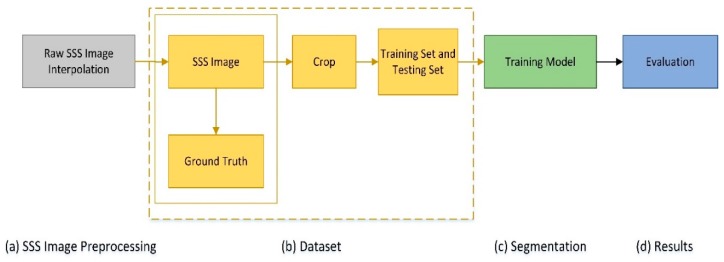
The experimental scheme of our proposed SSS image segmentation method.

**Figure 4 sensors-19-02009-f004:**
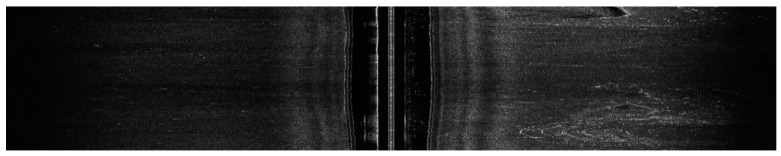
Examples of the raw SSS images.

**Figure 5 sensors-19-02009-f005:**
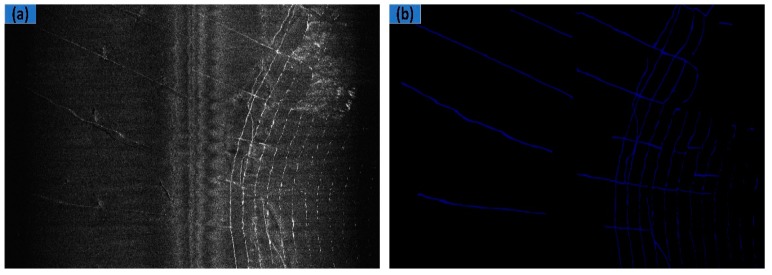
Examples of our data: (**a**) The results of the interpolation and removal operation used to form dataset; (**b**) corresponding ground truth.

**Figure 6 sensors-19-02009-f006:**
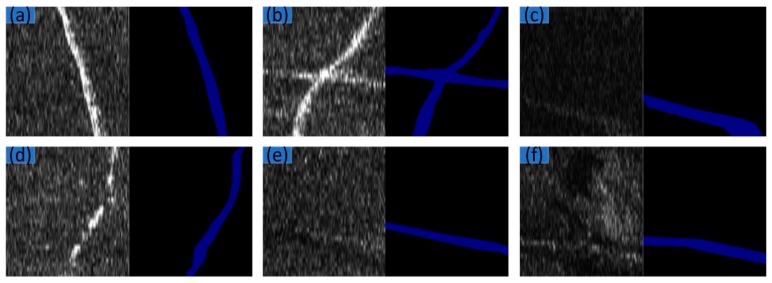
Examples of SSS image: (**a**) Clear and standard; (**b**) some target pixels are obvious and others are weak; (**c**) dark; (**d**) discontinuous; (**e**) weak and unclear; (**f**) noises.

**Figure 7 sensors-19-02009-f007:**
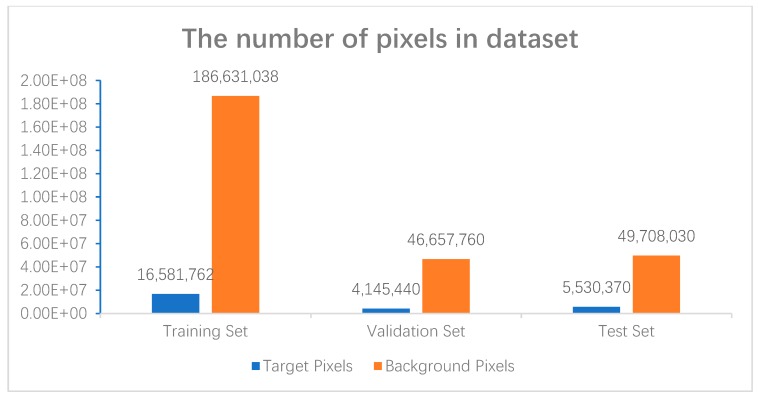
The number of pixels in dataset.

**Figure 8 sensors-19-02009-f008:**
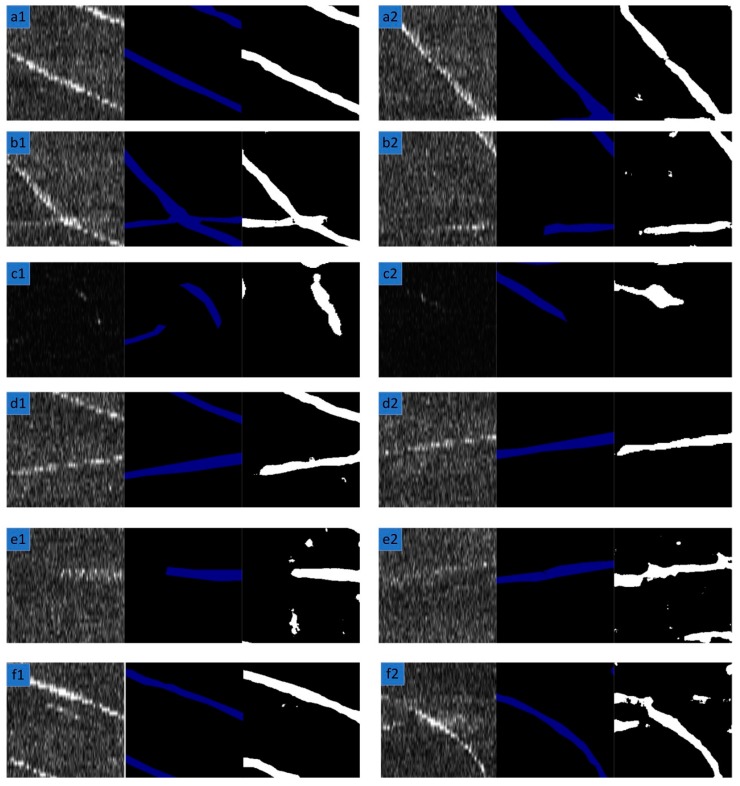
The test results on different data in the test set. From left to right, the images are the original SSS images, their corresponding ground truth and prediction results, respectively. Row1 (a1 and a2): clear and standard. Row2 (b1 and b2): some target pixels are obvious and others are weak. Row3 (c1 and c2): dark. Row4 (d1 and d2): discontinuous. Row5 (e1 and e2): weak and unclear. Row6 (f1 and f2): strong noise.

**Table 1 sensors-19-02009-t001:** The number of input and output feature maps.

Block	Encoder		Decoder
	A	b	
1	3	64	32
2	64	128	64
3	128	256	128
4	256	512	256

**Table 2 sensors-19-02009-t002:** The receptive field (rf) and stride size (s_size) in encoder module used in ECNet.

Block	Encoder1	Encoder2	Encoder3	Encoder4
rf	8	22	50	106
s_size	2	4	8	16

**Table 3 sensors-19-02009-t003:** Comparison of different decoding methods in our architecture. (The best results are bold)

Decoding Method	Pixel acc. (%)	Mean acc. (%)	Mean IU (%)	f.w. IU (%)
U-Net	91.09	75.75	64.71	85.38
SegNet	90.79	73.57	63.21	84.86
LinkNet	91.01	74.40	63.94	85.17
Ours	**91.62**	**77.05**	**66.18**	**86.10**

**Table 4 sensors-19-02009-t004:** Comparison of different architectures performance in dataset. (The best results are bold)

Model	Pixel acc. (%)	Mean acc. (%)	Mean IU (%)	f.w. IU (%)
U-Net	**92.68**	75.02	**67.13**	**87.23**
SegNet	91.99	73.69	65.22	86.29
LinkNet	91.45	73.63	64.28	85.64
Ours	91.62	**77.05**	66.18	86.10

**Table 5 sensors-19-02009-t005:** Comparison of different architectures implementation on NVIDIA Xavier. (The best results are bold)

Model	Parameters	Model Size	Time
U-Net	31.0 M	124.1 MB	87.6 ms
SegNet	28.4 M	113.8 MB	93.9 ms
LinkNet	21.6 M	86.7 MB	29.4 ms
Ours	**4.7 M**	**18.7 MB**	**27.9 ms**

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
