# Peer review of "ECNet: Efficient Convolutional Networks for Side Scan Sonar Image Segmentation"

_sensors, 2019, doi:10.3390/s19092009_

Round 1

Reviewer 1 Report

Expansion of IU?

Give the inference from validation measures.

The paragraph in Page.9 (lines 266-279)  may be included in methodology section.

Give reasons why U -Net has achieved more accuracy and IU in table 4? Why the proposed method has resulted in lesser values.

How does the proposed method handle speckle noise and intensity inhomogeneity?

Expand HED

Advantange of using bilinear interpolation in pre-processing

What is the range of hyperparameters used in ECNET and how to arrive at optimal values for the parameters?

Line number 281 - it if figure 3(b) not (a)

Author Response

Manuscript ID: sensors-478706
Type of manuscript: Article
Title: ECNet: Efficient Convolutional Networks for Side Scan Sonar Image
Segmentation
Authors: Meihan Wu, Qi Wang, Eric Rigall, Kaige Li, Wenbo Zhu, Bo He *,
Tianhong Yan *
Received: 21 March 2019

Thank you very much for your comments and suggestions. Those comments are very helpful for revising and improving our paper. We have studied the comments carefully and have made response to it. Our answers are written in the word file. We have tried our best to improve the manuscript and all the revisions have been highlighted by yellow color in the manuscript. Hope this can help you to trace the modifications easily.

Reviewer 2 Report

The paper presents a novel convolutional neural network architecture, ECNet, for side scan sonar (SSS) image segmentation. The authors combine the encoder-decoder structure from SegNet, the direct connection from LinkNet and the side-output structure from holistically-nested edge detection (HED) method to form the architecture of ECNet. Specifically, ECNet contains four pairs of encoder and decoder blocks. The encoder blocks extract feature maps of the image in different scales. Same as SegNet, the decoder network uses the max-pooling indices from the corresponding encoder blocks to perform nonlinear up-sampling for obtaining a sparse feature map. Finally, the side-outputs from the first three encoder blocks and the main-outputs from the first decoder block are used for pixel-wise segmentation of SSS images. The parameters of the network are learned via backpropagation and stochastic gradient descent. The performance of the proposed network architecture is tested on real SSS images and measured by pixel accuracy, mean accuracy, as well as the mean and frequency weighted intersection-over-union scores. Comparing to U-Net, SegNet, and LinkNet, the proposed ECNet achieves a similar level of segmentation performance with much less network complexity and inference time for SSS image segmentation.

The results shown in the paper are generally compelling. However, the performances of the ECNet are only benchmarked on one set of SSS images with one type of targets and seabed textures. It will be better to show the generality of the proposed method by testing its segmentation performances with various settings, including different types of targets and different seabed textures. Moreover, since none of the U-Net, SegNet, and LinkNet are designed for sonar image segmentation, it will be more useful to add benchmarks between ECNet and other SSS segmentation algorithms.

Specific comments:

The authors compare the performances of the decoding methods in Table 3. However, the experimental procedure is not explicitly written in the context.

At line 251, it seems that the authors select the model that performs the best on their testing set. However, the authors should obtain the model that performs best on the validation set and evaluate it on the testing set. This might be a typo since the authors mention having a validation set at line 225.

At line 300, the authors claim that their experiments prove that reducing the receptive field has a small impact on segmenting the sonar images. I suggest including those experiment results and/or some related analysis since this may be a disadvantage of the ECNet. 

I also recommend adding more references to SSS image classification and segmentation works including:

Zhu, P., Isaacs, J., Fu, B., & Ferrari, S. (2017, December). Deep learning feature extraction for target recognition and classification in underwater sonar images. In 2017 IEEE 56th Annual Conference on Decision and Control (CDC) (pp. 2724-2731). IEEE.

Chang, S., Isaacs, J., Fu, B., Shin, J., Zhu, P., & Ferrari, S. (2018, April). Confidence level estimation in multi-target classification problems. In Detection and Sensing of Mines, Explosive Objects, and Obscured Targets XXIII (Vol. 10628, p. 1062818). International Society for Optics and Photonics.

Celik, T., & Tjahjadi, T. (2011). A novel method for sidescan sonar image segmentation. IEEE Journal of Oceanic Engineering, 36(2), 186-194.

Author Response

Manuscript ID: sensors-478706
Type of manuscript: Article
Title: ECNet: Efficient Convolutional Networks for Side Scan Sonar Image 
Segmentation
Authors: Meihan Wu, Qi Wang, Eric Rigall, Kaige Li, Wenbo Zhu, Bo He *, 
Tianhong Yan *
Received: 21 March 2019

Thank you very much for your comments and suggestions. Those comments are very helpful for revising and improving our paper. We have studied the comments carefully and have made response to it. Our answers are written in the Word file. We have tried our best to improve the manuscript and all the revisions have been highlighted by yellow color in the manuscript. Hope this can help you to trace the modifications easily.

Round 2

Reviewer 1 Report

Avoid the use of  words such as "WE'' in the revised version

Accuracy values need to addressed in percentage

Diagrams could be avoided in related works section

Author Response

Thank you very much for your comments and suggestions. Those comments are very helpful for revising and improving our paper. We have studied the comments carefully and have made response to it. Our answers are written in the Word file. 

Reviewer 2 Report

The clarity of the content was significantly improved in the revised manuscript. The author answered all my specific questions well. However, one of my previous concern that the ECNet was only tested with one dataset was not addressed in the revision. The current content of the manuscript is good for publication, but I suggest the author add the test of the ECNet’s generality in their future works.   

Besides, there is a typo at line 326: "ECnet"

Author Response

(The authors gave the same response as above.)
